# Choosing important health outcomes for comparative effectiveness research: 5th annual update to a systematic review of core outcome sets for research

Elizabeth Gargon *, Sarah L. Gorst, Paula R. Williamson

Department of Biostatistics, University of Liverpool, Liverpool, United Kingdom

* e.gargon@liverpool.ac.uk

## Abstract

### Background

A systematic review of core outcome sets (COS) for research is updated annually to populate an online database. It is a resource intensive review to do annually but automation techniques have potential to aid the process. The production of guidance and standards in COS development means that there is now an expectation that COS are being developed and reported to a higher standard. This is the fifth update to the systematic review and will explore these issues.

### Methods

Searches were carried out to identify studies published or indexed in 2018. Automated screening methods were used to rank the citations in order of relevance. The cut-off for screening was set to the top 25% in ranked priority order, following development and validation of the algorithm. Studies were eligible for inclusion if they reported the development of a COS, regardless of any restrictions by age, health condition or setting. COS were assessed against each of the Core Outcome Set-STAndards for Development (COS-STAD).

### Results

Thirty studies describing the development of 44 COS were included in this update. Six COS (20%) were deemed to have met all 12 criteria representing the 11 minimum standards for COS development (range = 4 to 12 criteria, median = 10 criteria). All 30 COS studies met all four minimum standards for scope. Twenty-one (70%) COS met all three minimum standards for stakeholders. Twenty-three studies (77%) included patients with the condition or their representatives. The number of countries involved in the development of COS ranged from 1 to 39 (median = 10). Six studies (20%) met all four minimum standards [five criteria] for the consensus process.

**Data Availability Statement:** All relevant data are within the paper and its Supporting Information files.

**Funding:** The author(s) received no specific funding for this work. Professor Williamson is a National Institute for Health Research (NIHR) Senior Investigator. The views expressed in this article are those of the author(s) and not necessarily those of the NIHR, or the Department of Health and Social Care.

**Competing interests:** EG and PW are members of the COMET Management Group. All other authors had no competing interests. This does not alter our adherence to PLOS ONE policies on sharing data and materials.

## Conclusion

Automated ranking was successfully used to assist the screening process and reduce the workload of this systematic review update. With the provision of guidelines, COS are better reported and being developed to a higher standard.

## Introduction

Outcomes in research need to be relevant and important to patients and the public, health care professionals and other stakeholder groups involved in health care decision making. Choosing meaningful outcomes could improve patient's care, and if all studies in a particular area of healthcare measured and reported the same important outcomes, they could all be combined to better inform decisions about healthcare; thus reducing waste and thereby saving money. This is being achieved through the development and use of core outcome sets (COS). These sets represent the minimum that should be measured and reported in all clinical trials of a specific condition, and are suitable for use in clinical audit or research other than randomised trials [1]. COMET has successfully brought together the international COS literature in an online database (http://www.comet-initiative.org/studies/search). A systematic review of COS is updated annually to keep the database current [2–6].

Use of the COMET website continues to increase (source: Google Analytics), with 33,460 visits from 20,777 users in 2018, and a rise in the number of visits from outside the UK (19,696 visits; 59% of all visits). Inclusive of December 2018, 25,959 searches had been completed, with 6521 in 2018 alone (a 27% increase since 2017). The growing awareness of the need for COS is reflected not only in the website and database usage figures, but in the marked increase in the number of included COS in the last update of the systematic review [6]. A survey of COMET database users highlighted that the two most common reasons for searching the database was 'thinking about developing a COS' and 'planning a clinical trial', suggesting increased awareness of both the need for COS development and the use of COS [3].

The production of guidance and standards in the area of COS development means that not only are more COS being developed, but there is an expectation that they are being developed and reported to a higher standard [1, 7–9]. As methodological research in this area is still in its infancy, COMET guidance and standards will be updated in the future to reflect new evidence as it becomes available. Minimum standards for COS development have been established to improve the methods used in COS development, as well as to help users of COS to evaluate whether a COS has been developed using appropriate methodology [7]. A recent study assessed whether published cancer COS met the Core Outcome Set-STAndards for Development (COS-STAD) criteria [10]. No COS met all of the minimum standards, with most studies meeting half of the standards. However, it was acknowledged that COS-STAD was not published until 2017 which was after the cancer COS had all been initiated; therefore, this was a baseline against which future comparisons can be made.

The annual update to this systematic review is crucial to maintain a valuable repository of COS. However, it is a labour intensive review and therefore costly to do this annually. Challenges with this search have been previously documented, and include the absence of MeSH headings for COS, variability in text used to describe these types of studies, and the broad characteristics of this search [11]. The accumulation of these features result in the manual screening of a large number of records to identify a relatively small number that are relevant. Automation techniques have great potential to make systematic reviews quicker and cheaper;

and machine learning, using a logistic regression method, has been used to develop a system for ranking articles during the screening process of a systematic review [12]. The previous updates of this review have been used to train and evaluate this method with success, and the results suggest that the machine learning model could reduce the workload of this and future updates of the systematic review of COS, by up to 75% [13].

The aims of the current study were therefore to:

1. update the systematic review of COS to identify COS that were published or indexed in 2018;

2. test the automated screening approach prospectively, and

3. assess the methods of development, against minimum standards, of the included COS.

## Methods

### Systematic review update

The systematic review methods used in this update have been previously described [2–6]. A summary is provided here with expansion on new methods being used for this update.

### Study selection

Studies were eligible for inclusion if they had applied methodology for determining which outcomes (or outcome domains) should be measured in clinical trials or health research. As described previously, studies were eligible for inclusion if they reported the development of a COS, regardless of any restrictions by age, health condition or setting. The inclusion and exclusion criteria remain unchanged, and were described in full in the original systematic review and previous update [2, 6].

### Identification of relevant studies

MEDLINE via Ovid and SCOPUS were searched in March 2019, without language restrictions, to identify studies that had been published or indexed between (and inclusive of) January 2018 and December 2018. The search strategy, developed for the original review, was used for the current update [2] (S1 Table). 'MEDALL' was applied, as described in the previous update [6], to capture 'in-process' citations as well as E-pub ahead of print citations. Hand searching was completed, including studies that had been submitted to the COMET database/website, reference lists in eligible studies, as well as those in ineligible studies that referred to a COS.

### Selecting studies for inclusion in the review

As described previously [2, 5], records from each database were combined and duplicates removed.

Automated screening methods were used to rank the citations, in order of relevance, identified in this update [12, 13]. As per the evaluation of this model, the cut-off for screening was set to the top 25% of abstracts in ranked priority order [13]. The titles and abstracts of the top 25% ranked citations were screened to assess eligibility (stage 1). The ranked list was ordered alphabetically by author surname, prior to any screening, to avoid ranked order bias. The full text of potentially relevant articles were then assessed for inclusion (stage 2). Citations without an abstract were ineligible for ranking and therefore were all screened for eligibility.

Two reviewers independently screened the title and abstract of half of the citations each (EG and SG). Each citation was categorised as include, unsure, or exclude. Citations were assessed by a second reviewer (EG or SG) when there was any indecision; citations were discussed and categorised accordingly. If agreement was not achieved, the citation was referred to a third reviewer (PRW). Full papers were retrieved for all abstracts categorised as include or unsure.

Two reviewers independently assessed half of the full papers each (EG and SG) for inclusion in the review. As at abstract stage, indecisions at full paper assessment were discussed as necessary, and in cases of disagreement were referred to a third reviewer (PRW). The reasons for exclusion at this stage were documented for articles judged to be ineligible.

## Checking for correct exclusion

Full papers were obtained for a 1% sample of the records excluded on the basis of the title and abstract and checked for correct exclusion by a third reviewer (JK). If any studies were found to be incorrectly excluded, additional checking was performed within the other excluded records.

Of the records that had been excluded after reading the full text, 5% were assessed for correct exclusion (JK). If any studies were identified as being incorrectly excluded at this stage, further checking was performed.

## Assessment of COS-STAD minimum standards

One reviewer (EG or SG) independently assessed each of the COS against the COS-STAD criteria of development. As described in the assessment of cancer COS [10], a total of 12 criteria representing the 11 minimum standards were assessed in this study. The guidance on *how* to compare a published COS to the standards (Table 2 in a previous assessment of COS-STAD [10]) was used to aid assessment. Each criteria was assessed as yes (meeting that standard), no (not meeting that standard) or unsure (it was unclear whether the criteria had been met). A third reviewer (PRW) was consulted as necessary.

## Data extraction

As described in full previously [2], data was extracted by one reviewer (EG or SG) in relation to the study aim(s), health area, target population, interventions covered, methods of COS development and stakeholder groups involved. Text was extracted to support the COS-STAD assessment being made and to aid discussion where necessary.

## Prospective evaluation of the automated screening methods used to rank citations

We estimated the time taken for abstract and full paper screening based on one minute per abstract review, and 10 minutes per full paper review. We compared the full text screening stage against the last update to the systematic review (update 4 [6]).

We assessed a 1% sample of abstracts excluded by automated ranking to check for correct exclusion. If abstracts were found to be incorrectly excluded, further checking would be performed.

## Data analysis and presentation of results

The review is reported in accordance with PRISMA guidelines [14] (S1 PRISMA Checklist). Studies were described narratively, in text and tables. The median and range were presented to

summarise the number of the minimum standards met across all of the included COS studies. Percentage frequencies were used to report the number of COS that met each standard.

## Results

### Description of studies

Following the removal of duplicates, we identified 5878 records in the database search. Of these 5878 records, 239 had no abstract and so were ineligible for automated ranking. We used automated ranking for the remaining 5639 records and included the top 25% in order of relevance. All records ranked below the 25% cut off were excluded (n = 4229).

A total of 1649 records (1410 ranked and 239 no abstract) were manually screened. We excluded 1375 records during the title and abstract stage, and excluded a further 223 following the assessment of full text (Fig 1). The sample checking for correct exclusion of excluded citations matched 100%, with no records found to have been incorrectly excluded at any stage. Table 1 provides a summary of the reasons for exclusion of records at full text stage. Fifty-one records relating to 28 new studies met the inclusion criteria. The last included record was found at position 641/1410 ranked, and 1/239 no abstract records were included in this update (this was a linked report not a new study).

By using the automated screening methods to rank citations, we screened 75% less abstracts (1649/5878 abstracts), and consequently screened 50% less full texts than in the previous update (274 compared to 514 in the previous update). We estimate this to have saved approximately 110 hours, with the screening taking only 73 hours. All abstracts excluded by automated ranking were found to be correctly excluded (Fig 1).

We identified two additional records, through additional ongoing existing database alerts, as being eligible for inclusion in the review. These two studies were not identified during the review search, as although they were published in 2018, they had not been indexed in the databases at the time we ran our search. A further eight reports were identified by hand searching references of included studies. In total, 61 reports relating to 30 new studies describing the development of 44 COS were included for the first time in this update (S2 Table).

All 30 new COS studies were assessed against the COS-STAD criteria. An overview of the minimum standard assessments is provided in Table 2, and by study in S3 Table. Six of the 30 COS in this update (20%) were deemed to have met all 12 criteria representing the 11 minimum standards for COS development (range = 4 to 12 criteria, median = 10 criteria).

### Included studies

**Year of publication.** The results for year of first publication of COS have been updated to include the 30 new studies found in this update (Fig 2). Of the 30 studies identified in this update, 29 were published in 2018 and one study was published in 2017.

**Scope of core outcome sets.** The scope of published COS studies is summarised in Table 3. This includes study aims, setting for intended use, population characteristics and intervention characteristics. All 30 COS studies met all four minimum standards for scope. Regarding the research or practice setting(s) in which the COS is to be applied *(Standard #1)*, 26/30 (87%) studies stated that the COS was intended for use in clinical research and 4/30 (13%) stated that the COS was intended for use in clinical research and routine clinical practice. All studies described the health condition covered by the COS *(Standard #2)*. Fig 3 displays the number of COS that have been developed in each disease category. The COS identified in this update were developed across a range of health areas, with gastroenterology, dermatology, and pregnancy and childbirth being the most common areas. Disease categories

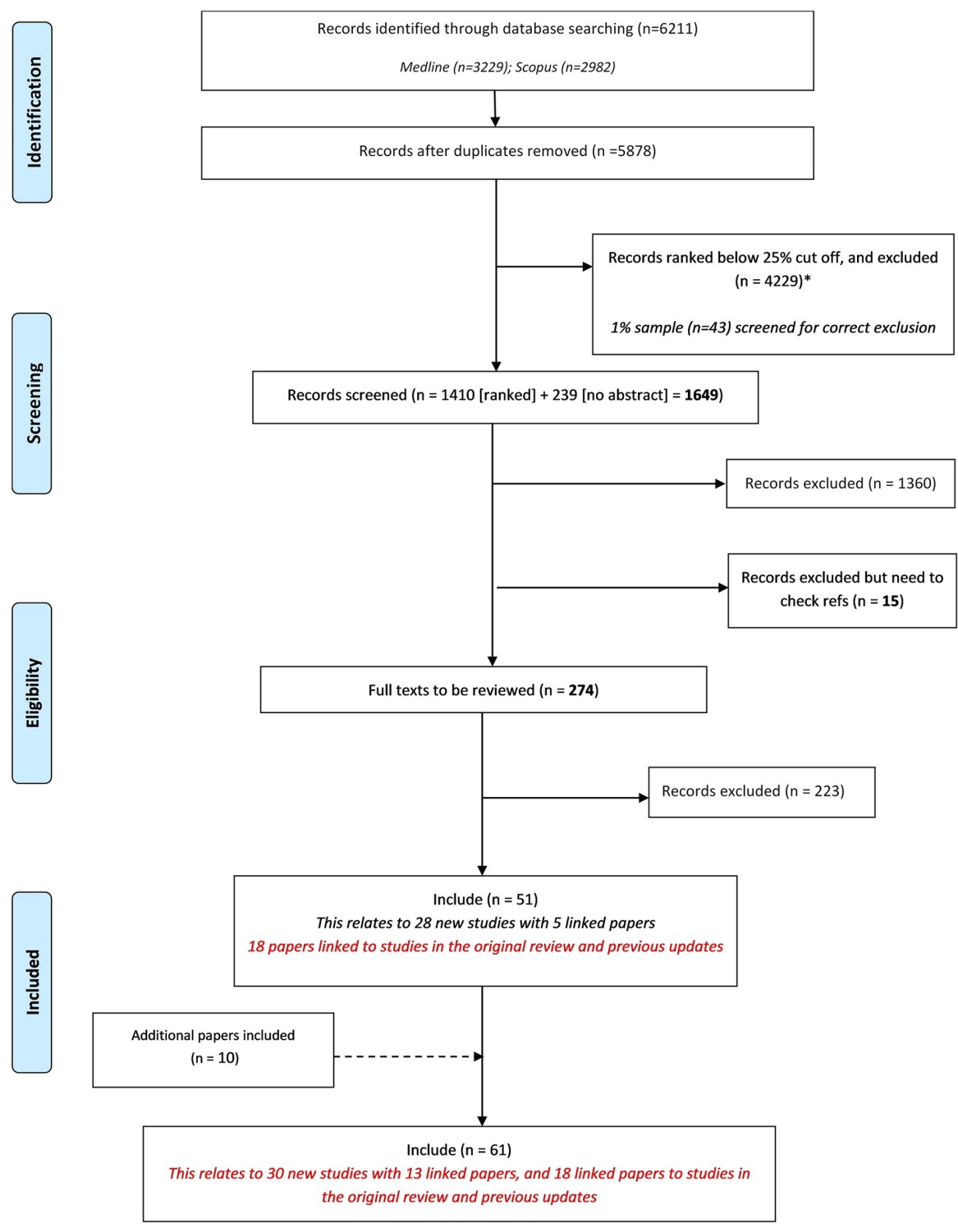

**Fig 1. PRISMA flowchart.**

**Table 1. Reasons for exclusion of records at full text stage.**

| Exclusion Categories of Full Text Stage | Number of records |
|---|---|
| Studies relating to how, rather than which, outcomes should be measured | 20 |
| Studies reporting the design/ rationale of single trial | 2 |
| Studies reporting the use of a COS | 1 |
| Systematic reviews of clinical trials | 1 |
| Review/overview/discussion only, no outcome recommendations | 80 |
| Core outcomes/ outcome recommendations not made | 32 |
| Quality indicators | 4 |
| One outcome/ domain only | 4 |
| Instrument development | 1 |
| Preclinical/ Early phase only (0, I, II) | 1 |
| Irrelevant | 33 |
| Assessed in previous review | 2 |
| HRQL | 1 |
| Recommendations for clinical management in practice not research | 19 |
| Registry development | 1 |
| Studies that elicit stakeholder group opinion regarding which outcome domains or outcomes are important | 10 |
| Ongoing studies | 11 |

and disease names are provided in S2 Table. All studies specified the population *(Standard #3)* and the intervention *(Standard #4)* covered by the COS (see Table 3).

**People involved in selecting outcomes.** Table 4 lists the stakeholders included in the development of the COS in this update and in the combined reviews. Twenty-one (70%) COS met all three minimum standards for stakeholders, that is that they included those who will use the COS in research, healthcare professionals and patients or their representatives. Three

**Table 2. COS minimum standards assessments summary (N = 30).**

| DOMAIN | STANDARD NUMBER | STANDARD | STANDARD MET N (%) | STANDARD UNCLEAR N (%) | STANDARD NOT MET N (%) |
|---|---|---|---|---|---|
| Scope specification | 1 | The research or practice setting(s) in which the COS is to be applied | 30 (100) | 0 | 0 |
| | 2 | The health condition(s) covered by the COS | 30 (100) | 0 | 0 |
| | 3 | The population(s) covered by the COS | 30 (100) | 0 | 0 |
| | 4 | The intervention(s) covered by the COS | 30 (100) | 0 | 0 |
| Stakeholders involved | 5 | Those who will use the COS in research | 24 (80) | 0 | 6 (20) |
| | 6 | Healthcare professionals with experience of patients with the condition | 27 (90) | 0 | 3 (10) |
| | 7 | Patients with the condition or their representatives | 23 (77) | 0 | 7 (23) |
| Consensus process | 8 | Initial list of outcomes considered both healthcare professionals' and patients' views | 16 (53) | 2 (7) | 12 (40) |
| | 9a | A scoring process was described a priori | 18 (60) | 9 (30) | 3 (10) |
| | 9b | A consensus definition was described a priori | 18 (60) | 9 (30) | 3 (10) |
| | 10 | Criteria for including/dropping/adding outcomes were described a priori | 14 (47) | 11 (37) | 5 (17) |
| | 11 | Care was taken to avoid ambiguity of language used in the list of outcomes | 13 (43) | 14 (47) | 3 (10) |

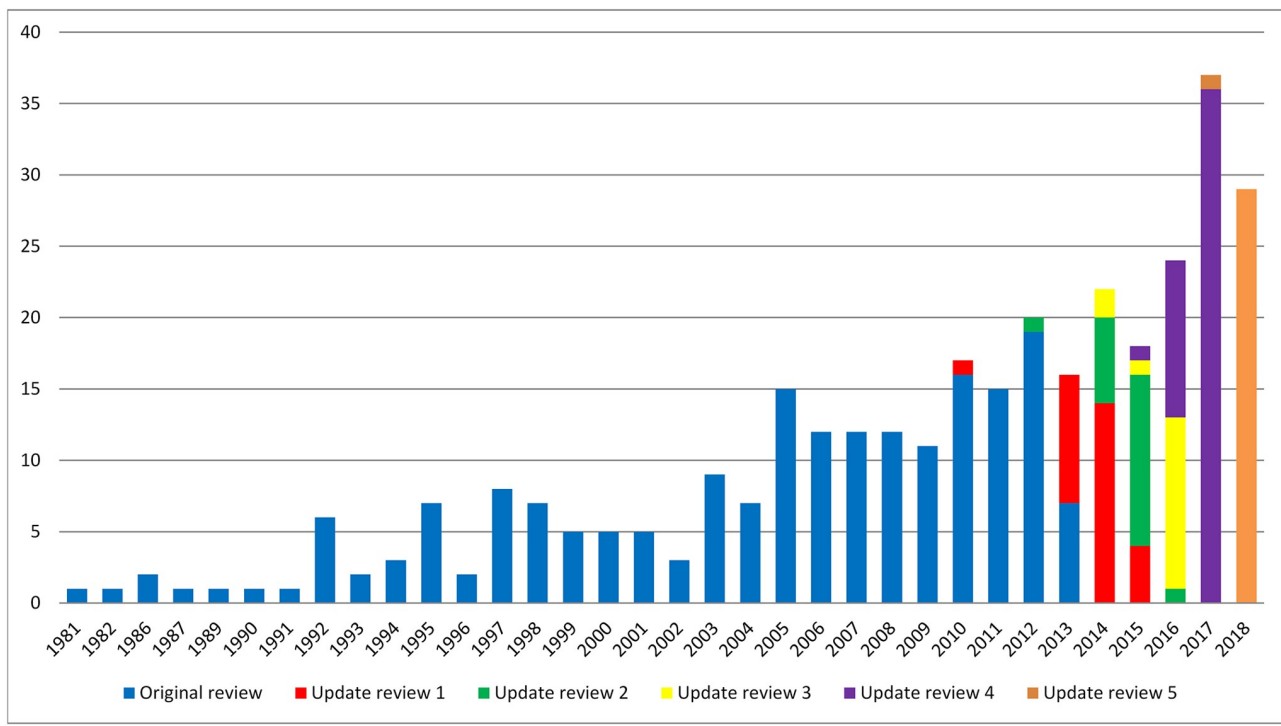

**Fig 2. Year of first publication of each COS study (n = 337).**

studies (10%) did not meet any of the stakeholder standards, as they consisted of a systematic review only on which core outcome recommendations were made, and therefore did not include participants directly in the development process. Twenty-four COS (80%) included those who will use the COS in research *(Standard #5)*. There were three instances where the stakeholders involved were clearly described and did not include those who will use the COS in research. Twenty-seven studies (90%) included healthcare professionals (HCPs) with experience of patients with the condition *(Standard #6)*. Twenty-three studies (77%) included patients with the condition or their representatives *(Standard #7)*. The remaining four studies clearly described stakeholders and did not include patients in the process.

Table 5 displays the 23 studies that reported details of public participation, highlighting that the Delphi and consensus meetings were the most commonly used methods with public representatives. The proportion of public participants, across all methods that included both clinical experts and the public, ranged from 4% [15] to 53% [16].

The geographical location of the participants included in COS development (Table 6) has been updated to include the COS identified in the current update. The number of countries involved in the development of the 30 new COS ranged from one to 39 (a median of 10). Twenty-six of the 30 studies (87%) included in the current review provided details on participant locations. Of these 26 studies, 25 (96%) included participants located in Europe, 19 (73%) in North America, 15 (58%) in Australasia, 10 (39%) in Asia, six (23%) in South America and six (23%) in Africa.

**Methods used to select outcomes.** The methods used in the 30 new COS studies identified in the current review are presented in Table 7 alongside the methods used in the five previous systematic reviews. Table 7 highlights that the majority of studies used mixed methods to

**Table 3. The scope of included studies (n = 337).**

| | Original review n (%) | Update review 1 n (%) | Update review 2 n (%) | Update review 3 n (%) | Update review 4 n (%) | Update review 5 n (%) | Combined* N (%) |
|---|---|---|---|---|---|---|---|
| **Study aims** | | | | | | | |
| Specifically considered outcome selection and measurement | 98 (50) | 21 (75) | 13 (65) | 10 (60) | 33 (69) | 29 (97) | 204 (61) |
| Considered outcomes while addressing wider clinical trial design issues | 98 (50) | 7 (25) | 7 (35) | 5 (40) | 15 (31) | 1 (3) | 133 (39) |
| **Intended use of recommendations** | | | | | | | |
| Clinical research | 176 (90) | 25 (89) | 19 (95) | 11 (73) | 44 (92) | 26 (87) | 301 (89) |
| Clinical research and practice | 20 (10) | 3 (11) | 1 (5) | 4 (27) | 4 (8) | 4 (13) | 36 (11) |
| **Population characteristics** | | | | | | | |
| Adults | 12 (6) | 12 (43) | 5 (25) | 10 (67) | 21 (44) | 17 (57) | 81 (24) |
| Children | 22 (11) | 2 (7) | 6 (30) | 0 (0) | 5 (10) | 4 (13) | 39 (12) |
| Adults and children | 12 (6) | 2 (7) | 0 (0) | 3 (20) | 10 (21) | 2 (7) | 30 (9) |
| Older adults | 2 (1) | 1 (4) | 0 (0) | 0 (0) | 3 (6) | 2 (7) | 8 (2) |
| Adolescents and adults | 0 (0) | 0 (0) | 0 (0) | 1 (7) | 4 (8) | 5 (17) | 10 (3) |
| Not specified | 148 (76) | 11 (39) | 9 (45) | 2 (13) | 5 (10) | 0 (0) | 169 (50) |
| **Intervention characteristics** | | | | | | | |
| All intervention types | 7 (4) | 8 (29) | 12 (60) | 8 (53) | 29 (60) | 17 (57) | 85 (25) |
| Drug treatments | 39 (20) | 4 (14) | 0 (0) | 0 (0) | 4 (8) | 4 (13) | 50 (15) |
| Surgery | 13 (7) | 4 (14) | 4 (20) | 4 (27) | 7 (15) | 3 (10) | 35 (10) |
| Vaccine | 2 (1) | 0 (0) | 0 (0) | 0 (0) | 0 | 0 | 2 (1) |
| Rehabilitation | 1 (1) | 1 (4) | 0 (0) | 1 (7) | 2 (4) | 0 | 5 (2) |
| Exercise | 1 (1) | 1 (4) | 1 (5) | 0 (0) | 1 (2) | 0 | 4 (1) |
| *Exercise (physical activity)* | *1* | *0* | *1* | *0* | *1* | *0* | |
| *Exercise (yoga)* | *0* | *1* | *0* | *0* | *0* | *0* | |
| Procedure | 4 (2) | 0 (0) | 2 (10) | 0 (0) | 2 (4) | 5 (17) | 13 (4) |
| Device | 3 (2) | 0 (0) | 0 (0) | 1 (7) | 0 | 0 | 4 (1) |
| Other | 11 (6) | 5 (18) | 0 (0) | 1 (7) | 3 (6) | 1 (3) | 21 (6) |
| Not specified | 115 (59) | 5 (18) | 1 (5) | 0 (0) | 0 | 0 | 118 (35) |

*Additional information provided by updated papers linked to previously published COS are reflected in the combined column

develop COS, with the use of the Delphi in combination with other methods being the most common (n = 23; 77%).

**Consensus process for agreeing outcomes.** Six studies (20%) met all four minimum standards [five criteria] for the consensus process.

Sixteen studies (53%) *considered both healthcare professionals and patients' views in the Initial list of outcomes* (Standard #8). Two studies (7%) did not clearly state whose views were considered when generating the initial list, and so therefore have been categorised as unclear. In one of these studies, a systematic review of reviews was conducted to inform the initial list, but it was not clear which study types were included in those reviews. In the remaining study, 'key informant' interviews were used to create an initial list of outcomes; no further description of the informants is provided, therefore we are unable to establish whose views were considered. Twelve studies (40%) did not meet this standard and did not consider both HCPs and patients' views when generating the initial list of outcomes used in the COS development. These studies considered trial data, clinical trials literature or clinical guidelines only (hence did not consider patients' views).

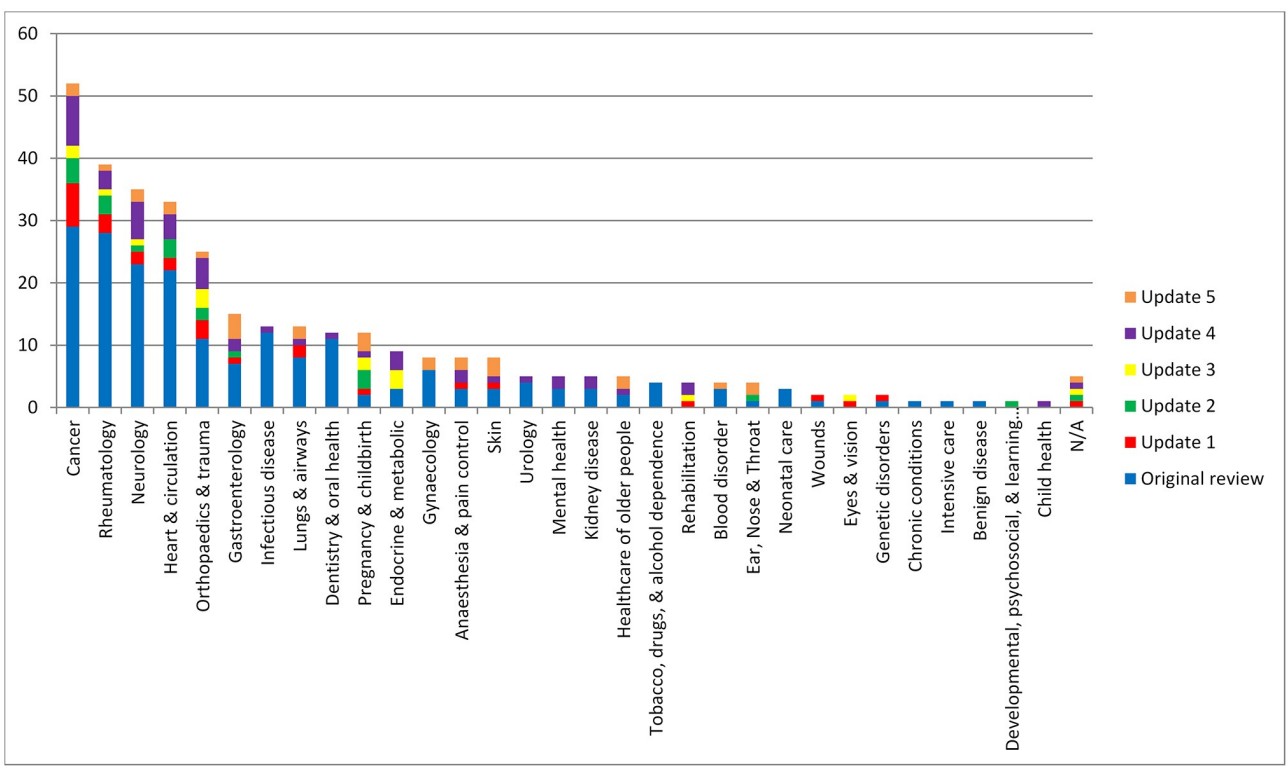

**Fig 3. Number of COS developed in each disease category (n = 337).**

Eighteen studies (60%) described there being both a scoring process and a consensus definition a priori, however it was unclear whether nine studies (30%) met this standard (Standard #9). In eight studies, it was unclear whether the scoring process and consensus definition were defined a priori, and one study did not describe specific methods relating to scoring or a process of consensus. Three studies (10%) did not meet this standard, as they were systematic reviews only, and therefore did not include a scoring process or consensus definition.

Fourteen studies (47%) described criteria for including/dropping/adding outcomes a priori (Standard #10). It was unclear whether this standard was met for 11 studies (37%). Detail was lacking in the description of this process for five studies, and for five studies that did describe these criteria, it was not possible to assess the a priori element. The criteria in this standard were described in the protocol of one study, but the reporting in the main paper conflicted with the protocol criteria and so it was deemed unclear whether the standard was met for this study. For the other five studies (17%) that did not meet this standard, three studies were systematic reviews only so did not include this process. One study only included one round of ranking outcomes and therefore did not include the process of including, adding and dropping outcomes. In the remaining study, criteria were decided after the first round of voting and therefore were not described a priori.

Thirteen studies (43%) took care to avoid ambiguity of language used in the list of outcomes (Standard #11). In eleven of these studies, consideration was given to ambiguity of language from the patient's perspective. In two studies, the questionnaire was piloted to assess usability for researchers and health care professionals. For 14 studies, (47%) there was no evidence that care was taken to avoid ambiguity of language, and therefore it was unclear whether they met

**Table 4. Participant groups involved in selecting outcomes for inclusion in COS (n = 337).**

| Participants category | Sub-category (not mutually exclusive) | Frequency of participants | | | | | | |
|---|---|---|---|---|---|---|---|---|
| | | Original review n | Update review 1 n | Update review 2 n | Update review 3 n | Update review 4 n | Update review 5 n | Combined |
| **Clinical experts** | | **171/172** | **20/21** | **17/17** | **13/14** | **41/43** | **27/27** | **289/294** |
| | Clinical experts | 86 | 14 | 16 | 13 | 36 | 27 | 192 |
| | Clinical research expertise | 66 | 9 | 9 | 2 | 23 | 13 | 122 |
| | Clinical trialists/ Members of a clinical trial network | 9 | 2 | | | 1 | | 12 |
| | Others with assumptions* | 54 | | | | | | 54 |
| **Public representatives** | | **30/172** | **13/21** | **11/17** | **8/14** | **27/43** | **23/27** | **112/294** |
| | Patients | 18 | 11 | 7 | 8 | 18 | 17 | 79 |
| | Carers | 7 | 1 | 3 | 3 | 8 | 8 | 30 |
| | Patient support group representatives | 9 | 1 | 4 | | 9 | 5 | 28 |
| | Service users | 2 | | | 1 | 2 | 1 | 6 |
| **Non-clinical research experts** | | **53/172** | **9/21** | **9/17** | **2/14** | **26/43** | **15/27** | **114/294** |
| | Researchers | 26 | 4 | 4 | 2 | 23 | 12 | 71 |
| | Statisticians | 19 | 4 | 3 | | 1 | | 27 |
| | Epidemiologists | 11 | 2 | 1 | | 4 | 3 | 21 |
| | Academic research representatives | 4 | | | | | 1 | 5 |
| | Methodologists | 6 | 3 | 2 | | 4 | 5 | 20 |
| | Economists | 3 | | 1 | | 2 | 1 | 7 |
| **Authorities** | | **39/172** | **5/21** | **3/17** | **0/14** | **12/43** | **5/27** | **64/294** |
| | Regulatory agency representatives | 30 | 4 | 3 | | 6 | 3 | 46 |
| | Governmental agencies | 12 | 1 | | | 5 | 1 | 19 |
| | Policy makers | 4 | 1 | | | 3 | 3 | 11 |
| | Charities | 1 | | | | 1 | 1 | 3 |
| | Service commissioners | | | | | 3 | 1 | 4 |
| **Industry representatives** | | **31/172** | **4/21** | **3/17** | **0/14** | **9/43** | **5/27** | **52/294** |
| | Pharmaceutical industry representatives | 28 | 3 | 3 | | 8 | 5 | 47 |
| | Device manufacturers | 2 | 1 | | | 1 | 1 | 5 |
| | Biotechnology company representatives | 1 | | | | | | 1 |
| **Others** | | **72/172** | **2/21** | **1/17** | **1/14** | **8/43** | **4/27** | **88/294** |
| | Service providers | | | | | 4 | | 4 |
| | Ethicists | 1 | | | | | | 1 |
| | Journal editors | 2 | | 1 | | 2 | 2 | 7 |
| | Funding bodies | | 1 | | | | 2 | 3 |
| | Yoga therapists/ instructors | | 1 | | | | | 1 |
| | Members of health care transition research consortium | | | | 1 | | | 1 |
| | Educationalist | | | | | 1 | | 1 |
| | Nutritionist | | | | | 1 | | 1 |
| | National professional and academic bodies/ committees | | | | | 1 | | 1 |
| | Guideline organisations | | | | | 1 | | 1 |
| | Others** (besides known participants) | 15 | | | | | | 15 |
| | Others with assumptions* | 54 | | | | | | 54 |
| **No details given** | | **24/196** | **7/28** | **3/20** | **1/15** | **5/48** | **3/30** | **43/337** |
| Not reported | | 13 | | | | | | 13 |

*(Continued)*

**Table 4.** (*Continued*)

| Participants category | Sub-category (not mutually exclusive) | Frequency of participants | | | | | | |
|---|---|---|---|---|---|---|---|---|
| | | Original review n | Update review 1 n | Update review 2 n | Update review 3 n | Update review 4 n | Update review 5 n | Combined |
| No participants | | 11 | 7 | 3 | 1 | 5 | 3 | 30 |

* 54 studies with clinical input but unclear about involvement of other stakeholders

** Workshop/meeting participants (*5), subcommittee/committee (*2), guidelines panel, military personnel, moderator and audience, representatives from EORTC, members with expertise in information technologies, informatics, clinical registries, data-standards development, expertise in vaccine safety, malaria control and representatives from funding agencies/registration authorities, and donor organisation, members of the Rheumatology Section of the American Academy of Pediatrics, the Pediatric Section of the ACR, and the Arthritis Foundation, the diagnostic radiology and basic science communities, and from individuals conversant with functional and quality of life (QOL) assessments, comparative effectiveness research, and cost/ benefit analysis

this criteria. The three studies (10%) that were exclusively systematic reviews did not meet this criteria due to the methods used.

## Discussion

In the fifth update to the systematic review of core outcome sets for research, we have identified 30 new studies describing the development of 44 COS. The annual publication of COS remains consistently high [6]. Their inclusion in the COMET database (http://www.comet-initiative.org/studies/search) brings the overall total of published COS to 337, relating to 410 COS. The annual completion of this review ensures that the COMET database is kept up-to date. We continually update the database as we identify eligible studies throughout the year, so were already aware of 22 of the included studies in this review. However, the systematic review has identified eight (27% of included) new COS studies that we had not previously identified through other methods, therefore underpinning the need to complete this annual update to the review to maximise the identification of COS for the COMET database. Use of automated screening methods to rank citations in this update to the systematic review considerably reduced the workload, time, and therefore cost, associated with this annual update. Furthermore, prospective use of this method has further validated the model for use in these updates [13] and this streamlining is promising for the necessary future updates to this review.

One fifth of the included COS in this update met all of the minimum standards for COS development, compared to no studies in the only other assessment of COS (in cancer) completed to date [10]. This improvement demonstrates that COS are being developed to a higher standard. The median number of criteria being met increased from six in the cancer COS assessment to 10 in those included in this update. In this assessment of minimum standards, fewer assumptions had to be made for all criteria, suggesting better reporting. In the previous update to the systematic review of COS it was reported that 21% of COS published in 2017, referenced the COS development reporting guidelines that had been published in 2016 [6, 8]. Eighteen (60%) of studies in this update referenced the reporting guidelines, demonstrating that COS are not only being developed to a higher standard but also reported to a higher standard. While our assessment was made on whether there was evidence in the paper(s) that COS-STAD criteria were met, other considerations would be necessary when assessing the applicability or usefulness of a COS; this would need to be evaluated *by the user* for the *intended purpose of use*.

The percentage of COS studies including all three relevant stakeholder groups has increased from 16% of studies in the cancer assessment of minimum standards, to 70% of studies in this

**Table 5. Nature of patient participation where detail is reported (n = 23).**

| | Methods used | Total number of participants | Number of public participants | % Public participants when multiple stakeholder groups included |
|---|---|---|---|---|
| **Allin** | Delphi | R1: 102 | R1: 42 | 41% |
| | | R2: 85 | R2: 31 | 37% |
| | | R3: 71 | R3: 22 | 31% |
| | Consensus meeting | 24 | 5 | 21% |
| | Measurement meeting | 14 | 1 | 7% |
| **Beuscart** | Interviews* | 15 | 15 | |
| | Delphi | R1: 150 | R1: 55 | 37% |
| | | R2: 136 | R2: 49 | 36% |
| | | R3: 129 | R3: 46 | 36% |
| | Consensus meeting* | 6 | 6 | |
| | Conference call^ | 6 | 0 | |
| | Conference call^^ | 3 | 0 | |
| **Callis Duffin** | Delphi | R1: 107 | R1:14 | 13% |
| | | R2: 77 | R2: 15 | 20% |
| | Stakeholder meeting | 88 | 8 | 9% |
| **Dos Santos** | Delphi | R1: 86 | R1:18 | 21% |
| | | R2: 71 | R2: 17 | 24% |
| | Consensus meeting | 20 | 3 | 15% |
| **Fish** | Interviews* | 19 | 19 | |
| | Delphi | R1: 149 | R1: 55 | 37% |
| | | R2: 149 | R2: 55 | 37% |
| | Consensus meeting | 23 | 13 | 57% |
| **Hall** | Delphi | R1: 670 | R1: 358 | 53% |
| | | R2: 586 | R2: 305 | 52% |
| | | R3: 533 | R3: 272 | 51% |
| | Consensus meeting | 54 | 26 | 48% |
| **Haywood** | Interviews* | 11 | 11 | |
| | Delphi | 168 | 69 | 41% |
| | Expert panel meeting | 23 | 4 | 17% |
| **Hopkins** | Delphi | R1: 114 | R1: 19 | 17% |
| | | R2: 67 | R2: not reported | |
| **Horbach** | Interview* | 2 | 2 | |
| | Delphi | R1: 317 | R1: 150 | 47% |
| | | R2: 257 | R2: 124 | 48% |
| | | R3: 244 | R3: 117 | 48% |
| | Consensus meeting | 39 | 8 | 21% |
| **Iorio** | Delphi and consensus meeting | 49 | 5 | 10% |
| **Kaiser** | Survey, Meeting, Delphi, Consensus meeting | 25 | 5 | 20% |
| **McGrattan** | Interviews | 63 | 33 | 52% |
| **Meher** | Delphi (prevention) | R1: 205 | R1: 30 | 15% |
| | | R2: 152 | R2: 22 | 14% |
| | Delphi (treatment) | R1: 197 | R1: 28 | 14% |
| | | R2: 143 | R2: 21 | 15% |
| | Meeting | 25 | 4 | 16 |
| **Murugupillai** | Delphi* | R1: 65 | R1: 65 | |
| | | R2: 42 | R2: 42 | |
| | Delphi^ | R1: 32 | R1: 0 | |
| | | R1: 29 | R1: 0 | |

*(Continued)*

**Table 5.** (Continued)

| | Methods used | Total number of participants | Number of public participants | % Public participants when multiple stakeholder groups included |
|---|---|---|---|---|
| **O'Donnell** | Delphi^ | R1: 242 | R1: 10 | 4% |
| | | R2: 189 | R2: 10 | 5% |
| | | R3: 169 | R3: 10 | 6% |
| | Meetings/webinar | 27 | 4 | 15% |
| Radner | Survey | 90 | 16 | 18 |
| | First meeting | 21 | not reported | not reported |
| | Online ratification | 23 | not reported | not reported |
| | Second meeting | 17 | not reported | not reported |
| Rankin | Delphi | R1: 152 | R1: 41 | 27% |
| | | R2: 148 | R2: 38 | 26% |
| | | R3: 127 | R3: 35 | 28% |
| Sahnan | Interviews | 21 | 21 | 100% |
| | Delphi | R1: 187 | R1: 66 | 35% |
| | | R2: 176 | R2: 57 | 32% |
| | | R3: 183 | R3: 59 | 32% |
| | Meeting | 47 | 14 | 30% |
| Singendonk | Delphi^ | R1: 125 | R1: 0 | |
| | | R2: 83 | R2: 0 | |
| | Delphi* | R1: 139 | R1: 139 | |
| | | R2: 127 | R2: 127 | |
| | Consensus meeting | not reported | not reported | not reported |
| Smith | Delphi | R1: 26 | R1: 5 | 19% |
| | | R2: 26 | R2: 5 | 19% |
| Spargo | Panel* | 11 | 11 | |
| | Delphi | R1: 82 | R1: 22 | 27% |
| | | R2: 78 | R2: 78 | 27% |
| | | R3: 74 | R3: 20 | 27% |
| Thorlacius | Interviews* | 42 | 42 | |
| | Delphi | R1: 93 | R1: 41 | 44% |
| | | R2: 86 | R2: 38 | 44% |
| | | R3: 83 | R3: 35 | 42% |
| | | R4: 79 | R4: 33 | 42% |
| | | R5: 78 | R5: 21 | 41% |
| | Consensus meeting 1 | 19 | 5 | 26% |
| | Consensus meeting 2 | 25 | 6 | 24% |
| Van den Bussche | Interviews* | 3 | 3 | |

*patient only

^clinician only

^^researcher only

update. Particularly, the participation of public participants in COS development continues to increase, with 71% of COS in this update including patients or their representatives compared to 56% in the last update [6]. All studies in this update that reported including public participants reported details about the rate of participation, again highlighting improved reporting of

**Table 6. Geographical locations of participants included in the development of each COS (n = 281).**

| Locations | Original review n (%) | Update review 1 n (%) | Update review 2 n (%) | Update review 3 n (%) | Update review 4 n (%) | Update review 5 n (%) | Combined* N (%) |
|---|---|---|---|---|---|---|---|
| North America | 134 (82) | 17 (68) | 9 (64) | 6 (55) | 28 (68) | 19 (73) | 215 (77) |
| Europe | 125 (76) | 19 (76) | 13 (93) | 10 (91) | 38 (93) | 25 (96) | 232 (83) |
| Australasia | 42 (26) | 4 (16) | 5 (36) | 3 (27) | 17 (41) | 15 (58) | 90 (32) |
| Asia | 34 (21) | 3 (12) | 6 (43) | 1 (9) | 18 (44) | 10 (39) | 75 (27) |
| South America | 16 (10) | 3 (12) | 2 (14) | 1 (9) | 13 (32) | 6 (23) | 43 (15) |
| Africa | 10 (6) | 1 (4) | 2 (14) | 1 (9) | 7 (17) | 6 (23) | 29 (10) |
| Total | 164 (84) | 25 (89) | 14 (70) | 11 (73) | 41 (85) | 26 (87) | 281 (83) |
| No details provided | 32 (16) | 3 (11) | 6 (30) | 4 (27) | 7 (15) | 4 (13) | 56 (17) |
| Median and range of number of countries | 6, 1–76 | 2, 1–33 | 6, 1–28 | 2, 1–18 | 6, 1–37 | 10, 1–39 | 4, 1–76 |

*Additional information provided by updated papers linked to previously published COS are reflected in the combined column

important COS development details. By including the relevant participants in COS development, COS are more likely to include the most relevant outcomes.

Participants from Europe and North America continue to be most prominent, but participation from other continents continues to increase. For example, over a third of COS here included participants from Asia, almost a quarter included participants from South America, and similarly for Africa. Furthermore, the median number of participant countries has increased from 6 in 2017 COS [6] to 10 in 2018 COS. This increase in multi-country COS

**Table 7. The methods used to develop COS (n = 337).**

| Main methods | Original review n (%) | Update review 1 n (%) | Update review 2 n (%) | Update review 3 n (%) | Update review 4 n (%) | Update review 5 | Combined* N (%) |
|---|---|---|---|---|---|---|---|
| **Semi-structured group discussion only** | 55 (28) | 2 (7) | 2 (10) | 0 (0) | 3 (6) | 0 (0) | 61 (18) |
| **Unstructured group discussion only** | 18 (9) | 0 (0) | 0 (0) | 0 (0) | 0 (0) | 0 (0) | 18 (5) |
| **Consensus development conference only** | 12 (6) | 0 (0) | 1 (5) | 0 (0) | 1 (2) | 0 (0) | 14 (4) |
| **Literature/systematic review only** | 11 (6) | 5 (18) | 2 (10) | 1 (7) | 6 (13) | 3 (10) | 28 (8) |
| **Delphi only** | 6 (3) | 2 (7) | 2 (10) | 0 (0) | 0 (0) | 2 (7) | 12 (4) |
| **Survey only** | 3 (2) | 0 (0) | 0 (0) | 0 (0) | 1 (2) | 0 (0) | 4 (1) |
| **NGT only** | 1 (1) | 0 (0) | 0 (0) | 0 (0) | 0 (0) | 0 (0) | 1 (<1) |
| **Interview only** | 0 (0) | 0 (0) | 0 (0) | 0 (0) | 1 (2) | 0 (00 | 1 (<1) |
| **Mixed methods** (*see descriptions below*) | 74 (38) | 17 (61) | 13 (65) | 12 (80) | 35 (73) | 25 (83) | 177 (53) |
| *Delphi + another method(s)* | 22 (11) | 6 (21) | 9 (45) | 9 (60) | 23 (48) | 23 (77) | 96 (29) |
| *Semi-structured group discussion + another method(s)* | 30 (15) | 7 (25) | 4 (20) | 2 (13) | 9 (19) | 2 (7) | 52 (15) |
| *Consensus development conference + another method(s)* | 7 (4) | 0 (0) | 0 (0) | 0 (0) | 1 (2) | 0 (0) | 8 (2) |
| *Literature/systematic review + another method(s)* | 10 (5) | 4 (14) | 0 (0) | 1 (7) | 2 (4) | 0 (0) | 17 (5) |
| *NGT + another method(s)* | 4 (2) | 0 (0) | 0 (0) | 0 (0) | 0 (0) | 0 (0) | 3 (1) |
| *Focus group + another method(s)* | 1 (1) | 0 (0) | 0 (0) | 0 (0) | 0 (0) | 0 (0) | 1 (<1) |
| No methods described | 16 (8) | 2 (7) | 0 (0) | 2 (13) | 1 (2) | 0 (0) | 21 (6) |

*Additional information provided by updated papers linked to previously published COS are reflected in the combined column

studies suggests that COS are becoming more international, but there remains a paucity in low and middle-income country (LMIC) participation and this remains an important area for improvement. Increased LMIC participation is necessary to increase global relevance and applicability of COS.

Previous updates have reported that the Delphi method is the popular choice for COS development. This update further supports that trend, with 83% of studies in this update utilising the Delphi technique, often in combination with other efforts. The minimum standards assessment of consensus process standards being met in this study was higher than had been previously reported in the cancer COS assessment of minimum standards, that being 20% of studies compared to 4% of cancer COS [10]. Though an improvement has been made, there are issues around the adequate descriptions of methods being provided in order to make an adequate assessment of the a priori status of consensus criteria. The recently published standards for COS protocol items (COS-STAP) [9], combined with the standards for development [7] and reporting [8], should help facilitate this much needed improvement.

There is a current interest in identifying how COS might fit into the different stages of the healthcare eco system. In this review we include COS for research, and over the years the percentage of COS for research that also intend their recommendations for use in routine care has remained constant at around 11% of COS. There is a growing interest in whether COS could serve a function all the way through the healthcare/research eco system. This raises questions about the methods used to develop COS for different purposes, and further work is warranted to investigate the current methods used for COS for various settings and methodological questions that need considering when multiple settings of use are intended under one set of COS recommendations. We have formed an informal international network of organisations with a shared interest in outcomes (COMET, ICHOM, CMTP, CS-COUSIN, OMERACT, McMaster University Grade Centre, COSMIN, CDISC), working towards sharing methodologies, and collaborating to deliver shared research activities to implement COS throughout the healthcare/research eco system.

In conclusion, we have completed the fifth update to a systematic review of COS for research, identifying 30 COS published and indexed in 2018. Automated ranking was successfully used to assist the screening process and reduce the workload of this systematic review update, while maintaining the accuracy. With the provision of guidelines, COS are better reported and being developed to a higher standard. While the time lag between guideline publication and the publication of the COS included in this review has been short, in time we expect improvements to continue to be made as COS developers become aware of the guidelines that now exist.

## Supporting information

**S1 PRISMA Checklist. PRISMA checklist for content of a systematic review.**
(DOC)

**S1 Table. Search strategy.**
(DOCX)

**S2 Table. Table of reports included in the updated review.**
(DOCX)

**S3 Table. COS minimum standards: Assessment by study (n = 30).**
(DOCX)

## Acknowledgments

We would like to acknowledge Jane Blazeby (Bristol University) and Mike Clarke (Queen's University Belfast) for their involvement in the conceptualisation and methodology in the original systematic review on which this update is based.

We would like to acknowledge Christopher Norman (University Paris-Saclay, University of Amsterdam) for running the automated screening model and applying it to the set of search results generated by the searches in this review, and generating the ranked list of citations used in this update.

We also acknowledge Jamie Kirkham (University of Manchester) for carrying out checking for correct exclusion at various stages of the review.

Professor Williamson is a National Institute for Health Research (NIHR) Senior Investigator. The views expressed in this article are those of the author(s) and not necessarily those of the NIHR, or the Department of Health and Social Care.

## Author Contributions

**Conceptualization:** Elizabeth Gargon, Paula R. Williamson.

**Data curation:** Elizabeth Gargon, Sarah L. Gorst.

**Formal analysis:** Elizabeth Gargon, Sarah L. Gorst.

**Funding acquisition:** Paula R. Williamson.

**Investigation:** Elizabeth Gargon, Sarah L. Gorst.

**Methodology:** Elizabeth Gargon, Paula R. Williamson.

**Writing – original draft:** Elizabeth Gargon, Sarah L. Gorst.

**Writing – review & editing:** Elizabeth Gargon, Sarah L. Gorst, Paula R. Williamson.

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
