## [Decision Letter · Decision Letter 0]

18 Oct 2019

PONE-D-19-22167

The use of automated article ranking in the fifth annual update to a systematic review of core outcome sets for research

PLOS ONE

Dear Dr Gargon,

Thank you for submitting your manuscript to PLOS ONE. After careful consideration, we feel that it has merit but does not fully meet PLOS ONE’s publication criteria as it currently stands. Therefore, we invite you to submit a revised version of the manuscript that addresses the points raised during the review process.

Reviewer 2 thought that the title could be misleading and I must say that I agree. Please make sure that the title is more focused on the content of the paper. For this specific reason he said that he was not willing review the manuscript that was outside his area of expertise. The two other reviewers were however positive and provided some (major and minor) issues that must be addressed before any final decision concerning your manuscript. 

We would appreciate receiving your revised manuscript by Dec 02 2019 11:59PM. To enhance the reproducibility of your results, we recommend that if applicable you deposit your laboratory protocols in protocols.io, where a protocol can be assigned its own identifier (DOI) such that it can be cited independently in the future. For instructions see: http://journals.plos.org/plosone/s/submission-guidelines#loc-laboratory-protocols

We look forward to receiving your revised manuscript.

Kind regards,

Florian Naudet, M.D., M.P.H., Ph.D.

Academic Editor

PLOS ONE

Journal Requirements:

'EG and PW are members of the COMET Management Group. SG has no competing interests.'

Please confirm that this does not alter your adherence to all PLOS ONE policies on sharing data and materials, by including the following statement: "This does not alter our adherence to  PLOS ONE policies on sharing data and materials.” (as detailed online in our guide for authors http://journals.plos.org/plosone/s/competing-interests).

If there are restrictions on sharing of data and/or materials, please state these. Please note that we cannot proceed with consideration of your article until this information has been declared.

Additional Editor Comments (if provided):

Reviewers' comments:

Reviewer's Responses to Questions

**Comments to the Author**

1. Is the manuscript technically sound, and do the data support the conclusions?

Reviewer #1: Partly

Reviewer #2: Partly

Reviewer #3: Yes

2. Has the statistical analysis been performed appropriately and rigorously? 

Reviewer #1: Yes

Reviewer #2: N/A

Reviewer #3: Yes

3. Have the authors made all data underlying the findings in their manuscript fully available?

Reviewer #1: Yes

Reviewer #2: No

Reviewer #3: Yes

4. Is the manuscript presented in an intelligible fashion and written in standard English?

Reviewer #1: Yes

Reviewer #2: No

Reviewer #3: Yes

5. Review Comments to the Author

Reviewer #1: Introduction : (minor comments)

Line 62 : the first sentence sounds like if « saving money » was the final goal of COS research. Is it ? Don’t you think that choosing meaningful outcomes could also improve patient’s condition ?

Line 73 : is this « awareness » only a COS developer awareness or also an awareness of COS users (trialists ? )

Line 75 : could you please cite any recent « guidance and standards » such as the COMET guidelines or more recently the revised OMERACT handbook. Do you think that these guidelines are the only method for COS development or is there any room for other methods ?

Line 86 : are you aware of any initiative to implement MeSH with headings for COS ? It would be very helpful

Line 98 : using an automated screenning approach is not an aim but a method. You already proved it worked in a previous paper cited ref 11.

Line 99 : using the COS-STAD is not an aim but a method also. The aim is to check if the included COS were developed with reasonable methods (as formulated by Kirkham 2017)

METHODS : (major comment)

Line 145 : Two comments to express that I doubt that only one researcher without medical background can assess the development of COSs for a large diversity of disease. The quality of assessment may be discussable.

Only one researcher assessed the included COS against COS Stad criteria. A double assessment of two researchers, one with an expertise in the given disease would be more transparent and relevant.

Moreover, the table 2 of the reference 8 has been made for cancer while supplementary data 3 show that cancer COSs were a minority. Did you adapt the criteria for each disease, for instance criteria 3 ? I doubt that it could be done without the advice of an expert of the disease.

DISCUSSION (minor comments)

Line 369 : It could be useful to have an online access to the assessment of each published tools on the COMET website for instance ie a table with the disease, the COS, and the COS-STAD assessment.

Reviewer #2: The title of the paper is misleading. No information about the algorithm is given in the paper. I am sorry, but without this information it does not make sense for me to review this paper. I strongly recommend to change the title of the paper.

Reviewer #3: The following manuscript reports an important and updated systematic review of COS published in 2018.

This review is an update of a yearly published systematic review, methods are well know and described.

Results are clear and comprehensive.

I only have a couple of comments on a single point:

- the authors aimed at assessing the use of an automated screening approach prospectively. Could the authors elaborate on this in the methods section?

- in the discussion, p21, lignes 364 and 366 : is there two different automated processes? from my understanding of the methods reported in the manuscript the authors used an automated screening method to rank the citations also reported as an automated ranking process.

6. PLOS authors have the option to publish the peer review history of their article (what does this mean?). If published, this will include your full peer review and any attached files.

Reviewer #1: No

Reviewer #2: No

Reviewer #3: Yes: Youri Yordanov

---

## [Author Response · Author response to Decision Letter 0]

29 Oct 2019

Reviewer #1: 

Introduction : (minor comments)

Line 62 : the first sentence sounds like if « saving money » was the final goal of COS research. Is it ? Don’t you think that choosing meaningful outcomes could also improve patient’s condition ?

Response: We agree that the inclusion of patient-important outcomes is one of the main benefits/features of COS, and have added an opening sentence into the introduction about the importance of outcomes being relevant to patients. 

Line 73 : is this « awareness » only a COS developer awareness or also an awareness of COS users (trialists ? )

Response: There is an increased awareness in both of these groups. We have added in a sentence, and a reference to a survey of COMET database users, that highlights this. 

Line 75 : could you please cite any recent « guidance and standards » such as the COMET guidelines or more recently the revised OMERACT handbook. Do you think that these guidelines are the only method for COS development or is there any room for other methods ?

Response: References to the COMET Handbook and standards have been added in, as well as a sentence to say that “As methodological research in this area is still in its infancy, COMET guidance and standards will be updated in the future to reflect new evidence as it becomes available.”

Line 86 : are you aware of any initiative to implement MeSH with headings for COS ? It would be very helpful

Response: Thanks for this comment. This is an ongoing discussion within the COMET Management Group. The introduction of a COS MeSH term in MEDLINE and other bibliographic databases for identifying COS papers would most definitely be very helpful. However, the inclusion of a new MeSH term in itself might be a challenge. MeSH is the National Library of Medicine’s controlled vocabulary thesaurus, who state that before a new descriptor is introduced, consideration must be given to how much is published about that topic; if little is published the Library see little purpose or advantage in creating a new descriptor in a vocabulary which has to encompass the subject content of the entire published literature. Although COS development is a growing area of research, it is relatively niche when compared to the broad nature of most medical subject headings. It is therefore probable that the proposal of a COS heading is unrealistic at this time. In the absence of a COS MeSH term, standards for reporting and development, as highlighted in the introduction to this paper, help with the searching and identification of COS studies. The use of standardised terms, for example ‘core outcome set’ in the title and abstract make it much easier to identify appropriate COS development papers.

Line 98 : using an automated screening approach is not an aim but a method. You already proved it worked in a previous paper cited ref 11.

Response: We have edited this to ‘test’ instead of use. 

Line 99 : using the COS-STAD is not an aim but a method also. The aim is to check if the included COS were developed with reasonable methods (as formulated by Kirkham 2017)

Response: Edit made. 

METHODS : (major comment)

Line 145 : Two comments to express that I doubt that only one researcher without medical background can assess the development of COSs for a large diversity of disease. The quality of assessment may be discussable.

Only one researcher assessed the included COS against COS Stad criteria. A double assessment of two researchers, one with an expertise in the given disease would be more transparent and relevant.

Response: Thank you for this comment. Our assessment was made on whether there was evidence in the paper(s) that the specific COS-STAD criteria were considered. This assessment does not require clinical expertise. However an assessment of the applicability or usefulness of a COS in a particular scenario would need to be evaluated by the user for the intended purpose of use. In this instance, it is likely that relevant clinical knowledge and understanding would be vital. We have added a comment about this into the discussion (see 2nd paragraph lines 401-404 in revised manuscript). 

Moreover, the table 2 of the reference 8 has been made for cancer while supplementary data 3 show that cancer COSs were a minority. Did you adapt the criteria for each disease, for instance criteria 3 ? I doubt that it could be done without the advice of an expert of the disease.

Response: Table 2 of the previous assessment of COS against COS-STAD was developed using cancer COS as an example, but not specifically for cancer COS. It was intended to facilitate users being able to compare a published COS to the standards. The paper states that consideration should be given to the specific disease area being considered (criteria 3), and that users need to use their own judgement regarding the scope for the purpose of use they require. Table 2 of the previous paper was only used to provide some guidance on how to compare the COS to the standards when needed, but the standards themselves were used for the assessment and the standards (COS-STAD) were developed to be applicable across diseases. 

We have edited this sentence to ‘remove the word ‘cancer’ to avoid any confusion. 

DISCUSSION (minor comments)

Line 369 : It could be useful to have an online access to the assessment of each published tools on the COMET website for instance ie a table with the disease, the COS, and the COS-STAD assessment.

Response: Thanks for this comment. We are currently considering this. While we recognise the potential merit of this, there is a need for individual consideration by the user of a COS for the specific intended purpose, therefore we have concerns that by including our assessment it could detract from people making their own critical assessment. However, we recognise this could be useful to some users, and are therefore considering it for our upcoming update to the COMET website and database, coming in 2020. 

Reviewer #2:

The title of the paper is misleading. No information about the algorithm is given in the paper. I am sorry, but without this information it does not make sense for me to review this paper. I strongly recommend to change the title of the paper.

Response: We have updated the title to be more in keeping with the previous updates to this systematic review: 

Choosing important health outcomes for comparative effectiveness research: 5th annual update to a systematic review of core outcome sets for research

Reviewer #3: 

The following manuscript reports an important and updated systematic review of COS published in 2018.

This review is an update of a yearly published systematic review, methods are well know and described.

Results are clear and comprehensive.

Response: Thank you for these comments.

I only have a couple of comments on a single point:

- the authors aimed at assessing the use of an automated screening approach prospectively. Could the authors elaborate on this in the methods section?

Response: Thank you for this comment. We have added in to the methods section and the results section the estimated time saved by using this method. 

- in the discussion, p21, lignes 364 and 366 : is there two different automated processes? from my understanding of the methods reported in the manuscript the authors used an automated screening method to rank the citations also reported as an automated ranking process.

Response: Your understanding is correct. We have edited this sentence to make this clearer.

---

## [Decision Letter · Decision Letter 1]

18 Nov 2019

Choosing important health outcomes for comparative effectiveness research: 5th annual update to a systematic review of core outcome sets for research

PONE-D-19-22167R1

Dear Dr. Gargon,

We are pleased to inform you that your manuscript has been judged scientifically suitable for publication and will be formally accepted for publication once it complies with all outstanding technical requirements.

I would like to thank you for the changes you made to the initial draft and the 2 reviewers for the insightful comments they made on the first draft. 

With kind regards,

Florian Naudet, M.D., M.P.H., Ph.D.

Academic Editor

PLOS ONE

Additional Editor Comments (optional):

Reviewers' comments:

Reviewer's Responses to Questions

**Comments to the Author**

1. If the authors have adequately addressed your comments raised in a previous round of review and you feel that this manuscript is now acceptable for publication, you may indicate that here to bypass the “Comments to the Author” section, enter your conflict of interest statement in the “Confidential to Editor” section, and submit your "Accept" recommendation.

Reviewer #1: All comments have been addressed

Reviewer #3: All comments have been addressed

2. Is the manuscript technically sound, and do the data support the conclusions?

Reviewer #1: Yes

Reviewer #3: Yes

3. Has the statistical analysis been performed appropriately and rigorously? 

Reviewer #1: Yes

Reviewer #3: Yes

4. Have the authors made all data underlying the findings in their manuscript fully available?

Reviewer #1: Yes

Reviewer #3: Yes

5. Is the manuscript presented in an intelligible fashion and written in standard English?

Reviewer #1: Yes

Reviewer #3: Yes

6. Review Comments to the Author

Reviewer #1: We thank the authors for having responded to all the comments. this paper will e useful for all researchers working on COS.

Reviewer #3: (No Response)

7. PLOS authors have the option to publish the peer review history of their article (what does this mean?). If published, this will include your full peer review and any attached files.

Reviewer #1: No

Reviewer #3: Yes: Youri Yordanov

---

## [Editor Report · Acceptance letter]

26 Nov 2019

PONE-D-19-22167R1 

Choosing important health outcomes for comparative effectiveness research: 5th annual update to a systematic review of core outcome sets for research 

Dear Dr. Gargon:

I am pleased to inform you that your manuscript has been deemed suitable for publication in PLOS ONE. Congratulations! Your manuscript is now with our production department. 

With kind regards,

on behalf of

Dr. Florian Naudet 

Academic Editor

PLOS ONE